# Physical Exercise Modulates miR-21-5p, miR-129-5p, miR-378-5p, and miR-188-5p Expression in Progenitor Cells Promoting Osteogenesis

**DOI:** 10.3390/cells8070742

**Published:** 2019-07-19

**Authors:** Maria Teresa Valenti, Michela Deiana, Samuele Cheri, Monica Dotta, Francesco Zamboni, Daniele Gabbiani, Federico Schena, Luca Dalle Carbonare, Monica Mottes

**Affiliations:** 1Department of Medicine, Internal Medicine, section D, University of Verona, 37134 Verona, Italy; 2Department of Neurosciences, Biomedicine and Movement Sciences, University of Verona, 37134 Verona, Italy

**Keywords:** physical exercise, miRNAs, RUNX2, PPARG2

## Abstract

Physical exercise is known to promote beneficial effects on overall health, counteracting risks related to degenerative diseases. MicroRNAs (miRNAs), short non-coding RNAs affecting the expression of a cell’s transcriptome, can be modulated by different stimuli. Yet, the molecular effects on osteogenic differentiation triggered by miRNAs upon physical exercise are not completely understood. In this study, we recruited 20 male amateur runners participating in a half marathon. Runners’ sera, collected before (PRE RUN) and after (POST RUN) the run, were added to cultured human mesenchymal stromal cells. We then investigated their effects on the modulation of selected miRNAs and the consequential effects on osteogenic differentiation. Our results showed an increased expression of miRNAs promoting osteogenic differentiation (miR-21-5p, miR-129-5p, and miR-378-5p) and a reduced expression of miRNAs involved in the adipogenic differentiation of progenitor cells (miR-188-5p). In addition, we observed the downregulation of PTEN and SMAD7 expression along with increased AKT/pAKT and SMAD4 protein levels in MSCs treated with POST RUN sera. The consequent upregulation of RUNX2 expression was also proven, highlighting the molecular mechanisms by which miR-21-5p promotes osteogenic differentiation. In conclusion, our work proposes novel data, which demonstrate how miRNAs may regulate the osteogenic commitment of progenitor cells in response to physical exercise.

## 1. Introduction

Bone formation and homeostasis are complex processes controlled by hormones, growth factors, signaling factors, and environmental factors. Bone homeostasis is achieved thanks to the cross-talk occurring between osteoblasts, osteocytes (mechanosensing osteoblasts embedded within the mineralized matrix), and osteoclasts (large multinucleated bone resorbing cells which differentiate from the monocyte/macrophage lineage). Osteoblasts derive from pluripotent bone marrow mesenchymal stromal cells (MSCs), which can differentiate into adipocytes, chondrocytes, or osteoblasts. Multiple signaling pathways control the mutually exclusive fates of progenitors by triggering the expression of master genes coding for specific transcription factors. Expression of the Runt-related transcription factor 2 (RUNX2) is essential for the osteogenic commitment; expression of a down-stream transcription factor, Osterix (SP7) allows osteoblastic maturation. Mature osteoblasts then express genes for matrix biosynthesis and mineralization, such as collagen type I genes (COL1A1/COL1A2), Osteocalcin (OCN), Osteopontin (SPP1), Osteonectin (SPARC), and others [1].

Notably, osteogenesis is regulated also at a post-transcriptional level. MicroRNAs (miRNAs) can interfere with mRNAs translation. Their role in controlling bone formation and homeostasis has been widely investigated [2]. In particular, it has been recognized that both specific transcription factors and miRNAs control mesenchymal cells commitment and differentiation, osteoblast and osteocyte functions, as well as osteoclast maturation [3]. Among miRNAs promoting osteogenic differentiation of progenitor cells, miR-21-5p, a ubiquitously expressed microRNA, plays an important role. One of miR-21 targets, Smad7, is an inhibitory member of the Smad proteins’ family, which antagonizes TGF-beta signaling [4,5]. Another player, miR-129-5p, promotes osteogenic differentiation of progenitor cells by targeting STAT 1, a transcriptional repressor of RUNX2 [6]. MiR-378 has been found to be overexpressed in BMP2-induced osteogenic differentiation of mesenchymal stromal cells [7]. MiR-188-5p, instead, seems to act as an important regulator of the mutually exclusive osteogenic/adipogenic commitment of progenitors. MiR-188-5p upregulation promotes PPARG expression, i.e., adipogenic differentiation [8].

Among the environmental factors influencing bone formation and quality, physical exercise certainly plays a fundamental role. Physical exercise enhances bone mineral density and can stimulate mesenchymal stromal cells mobilization and osteogenic commitment, to the exclusion of adipogenic commitment [9]. Recently, we have shown that the half marathon trial significantly enhances the expression of RUNX2 in amateur runners, as well as the calcification process in an in vitro model [10]. The present study was designed to further investigate the effects of the half marathon effort on MSCs alternative osteogenic/adipogenic commitment. For this purpose, cultures of a human bone marrow MSC line (hBMMSC) were supplemented with sera obtained from runners before (PRE RUN) and after (POST RUN) the marathon. MSC commitment and differentiation pathways were monitored by selected miRNAs’ expression analysis along with specific marker genes/proteins expression analysis.

## 2. Materials and Methods

### 2.1. Subjects

Twenty male runners were included in the study. Study subjects were enrolled during sport events called ‘Run For Science’, held in Verona (Italy) in April 2016 and April 2017. The twenty male amateur runners (median age 40.2 ± 8) carried out a 21.1 Km half marathon. Written informed consent was obtained from all participants and the study was approved by the Ethical Committee of Azienda Ospedaliera Universitaria Integrata of Verona, Italy (number 1538).

### 2.2. Sera Collection

Peripheral blood samples were collected before the run and immediately after. All participants gave informed consent. Sera were obtained from 10 mL of fresh blood by centrifugation at 400× *g*. Then, sera were harvested and frozen in aliquots at −80 °C until use.

### 2.3. In Vitro Treatments

Pooled sera were added to the Mesenchimal Stem Cell Grow Medium (PromoCell, GMBH Heidelberg, Germany) at 10% of final concentration. Cells were plated at density of 5 × 10^4^ cells per well into 24-well plates. The osteogenic differentiation was performed with osteogenic medium containing Osteogenic Stimulatory Supplements (15% Stemcell), 10^−8^ M dexamethasone, 3.5 mM β-glycerophosphate, and 50 μg/mL ascorbic acid (StemCell Technologies Inc, Vancouver, British Columbia, Canada) Adipogenic differentiation was performed by using isobutylmethylxanthine (0.5 mM), indomethacin (200 μM), dexamethasone (10^−6^ M), and insulin (10 μg/mL) in basal medium. For both osteogenic and adipogenic differentiation, the medium was changed every 3 days after initial plating.

### 2.4. Total RNA Extraction

Total RNA was extracted from each cell pellet using the RNA assay Minikit (Quiagen Italia, Milano, Italy) with DNAse I treatment. This kit provokes cell membrane breaks. Cell components obtained were then filtrated on QIAshredder columns, capable of binding nucleic acid. These can be then collected after washing and separation from other cell components. The amount of obtained RNA was quantified by measuring the absorbance at 260 nm. The purity of RNA was checked by calculating the ratio of the absorbance at 260 and 280 nm, with a ratio ranging from 1.8 to 2.0 taken to be pure.

### 2.5. Reverse Transcription

First-strand cDNA was generated, according to manufacturer’s protocol, using the First Strand cDNA Synthesis Kit (GE Healthcare, Italia, Milano, Italy) with random hexamers, reverse transcriptase, and 4 dNTPs. To RNA 1 μg, heated up to 65 °C for 10 min, kit components were added. Sample was then taken at 37 °C for 1 h. RT product was aliquoted in equal volumes and then stored at −80 °C.

### 2.6. Real Time RT-PCR

PCR was performed in a total volume of 50 μL containing 1x Taqman Universal PCR Master Mix, no AmpErase UNG, and 5 μL of cDNA from each sample. The following pre-designed, specific primers and probe set were obtained from Assay-on-Demand Gene Expression (Thermofisher Scientific Waltham, MA, USA) *RUNX2*, *hs00231692*m1; *PPARG,* hs01115513m1; *PTEN,* hs02621230m1; *SMAD7,* hs 00998193m1; *miR-21-5p*, TM.000397; *miR-188-5p*, TM 002320; *miR-129-5p*, TM 000590; *miR-378*-5p, TM.000567; *B2M*, hs999999m1, *GAPDH,* 0802021; *RNU*44, TM 001094. Real Time RT-PCR reactions were carried out in multiplex. The real time amplifications included 10 min at 95 °C, followed by 40 cycles at 95 °C for 15 s, and at 60 °C for 1 min. Thermocycling and signal detection were performed with ABI Prism 7300 Sequence Detector and signals were detected as we previously reported [11]. The expression levels were calculated for each sample in triplicate after normalization against the housekeeping genes (β₂ microglobulin and GADPH for mRNA or RNU44 for miRNAs), using the relative fold expression differences.

### 2.7. Western Blotting

Protein extraction was performed using Ripa buffer (Thermo Fisher Scientific, Waltham, MA, USA) according to the manufacturer’s protocol. Protein concentrations were calculated by performing a BCA assay (Thermo Scientific, Waltham, MA, USA). Protein samples were diluted in 4x Laemmli’s sample buffer (Biorad, CA, US), heated for 5 min at 95 °C, and separated by sodium dodecyl sulfate−polyacrylamide gel electrophoresis (SDS PAGE), using a mini-PROTEAN ^®^TGXTM Precast gradient 4–20% gel (Biorad, CA, US), followed by transfer onto polyvinylidene difluoride (PVDF) membranes (Thermo Fisher Scientific, Waltham, MA, USA). PVDF membranes were probed with the primary (anti-Caspase-3 antibody [EPR18297] (Abcam ab184787; Cambridge, UK) (AKT (C67E7), (Cell Signaling, 4691); pAKT (193H12), Cell Signaling, 4058; SMAD4 (B-8), (SantaCruz Biotech., Dallas, TX, USA; β ACTIN (BA3R), (Thermo Scientific, Waltham, MA, USA); and secondary antibodies (Anti-rabbit (Cell Signaling, 7074); Anti-mouse (Cell Signaling, 7076). Signals were detected using a chemiluminescence reagent (ECL, Millipore, Burlington, MA, USA) according to the manufacturer’s instructions. Images were acquired using a LAS4000 Digital Image Scanning System (GE Healthcare, Little Chalfont, UK). Densitometric analysis was performed by using ImageQuant software (GE Healthcare, Little Chalfont, UK), and the relative protein band intensity was normalized to β ACTIN as we previously reported [12].

### 2.8. Alizarin Red Staining

Calcium deposition was evaluated after 21 days of culture by the alizarin red staining as we previously reported [13]. Briefly, cells were fixed with 70% ethanol, washed with water, and stained for 5 min with 40 mM Alizarin Red S at pH 4.1. Then, the cells were rinsed with 1× phosphate-buffered saline.

### 2.9. Oil Red O Staining

Lipid droplets were stained using the Oil Red O, according to the manufacturer’s instructions. The total area of red pixels in the Oil Red O-stained droplets/cell was determined by using the IMAGE J image analysis as previously reported [13]. In particular, positively stained areas were expressed as percentage respect to total area. In addition, we calculated the number of lipid droplets for each cell as previously reported [14].

### 2.10. Statistic Analysis

Statistic analyses were performed using SPSS 21.0 for Windows operative system. To compare all variables we used Student T test, considering statistically relevant *p* values < 0.05. Results were expressed as Mean ± Standard Deviation. To evaluate any variable relation, we used bivariate correlation.

## 3. Results

### 3.1. Expression of miRNAs in Osteoprogenitor Cells

To evaluate the effects of physical exercise on osteo-miRNAs modulation, we analyzed the expression of miR-21-5p, miR-129-5p, miR-188-5p, and miR-378-5p, respectively, which are known to be involved in osteogenic differentiation and modulation [5,6,7,8]. As reported in Figure 1, expression levels of miR-21-5p were higher in MSCs treated with POST RUN sera compared to cells treated with PRE RUN sera after 7 (*p* < 0.01), 14 (*p* < 0.05), and 21 (*p* < 0.01) days of differentiation. Similarly, miR-129-5p expression levels were higher in MSCs treated with POST RUN sera after 14 days of differentiation (*p* < 0.01). No differences were observed after 7 or 21 days of differentiation. Conversely, miR-188-5p expression appeared significantly downregulated after 14 and 21 days of osteogenic differentiation (*p* < 0.05). MiR-378-5p was not detected after 7 days of differentiation in MSCs treated either with PRE RUN or POST RUN sera. It appeared to be upregulated after 14 (*p* < 0.05) and 21 (*p* < 0.05) days of differentiation in the presence of POST RUN sera.

### 3.2. Osteogenic Differentiation of Mesenchimal Stromal Cells

The enhanced osteogenic differentiation of MSCs treated with POST RUN sera suggested by miRNAs modulation was confirmed by the upregulation of RUNX2 expression (Figure 2A), and by the increased number of bone nodule formation evaluated by Alizarin Red staining (Figure 2B). Accordingly, upregulation of COL1A2 (collagen type I alpha 2 chain; Figure 2D) and of IBSP (bone sialoprotein 2; Figure 2C) gene expression, and increased OCN (Osteocalcin) protein levels (Figure 2E) confirm the production of bone ECM proteins.

Treatment of MSCs with POST RUN sera protected them from apoptosis. In fact, a lower cleaved caspase-3/caspase-3 ratio was observed in POST RUN compared to PRE RUN-sera treated MSCs (Figure 3A). The downregulation of PPARG2 expression (Figure 3B) and reduced oil droplets evaluated by the Oil Red O staining (Figure 3C), suggest reduced adipogenic commitment in MSCs treated with POST RUN sera.

### 3.3. MiR-21-5p Promotes Osteogenic Differentiation by Targeting PTEN and SMAD7 mRNAs

In order to further explore miR-21-5p’s action towards osteogenic differentiation, we analyzed the expression of PTEN and SMAD7 genes. Both PTEN (Figure 4A) and SMAD7 (Figure 4B) RNA levels were reduced in MSCs treated with POST RUN sera. In addition, AKT and pAKT (Figure 4C) as well as SMAD4 (Figure 4D) protein levels increased in MSCs treated with POST RUN sera.

The observed AKT protein increase, along with a significant reduction of PTEN mRNA levels, suggest that physical exercise promotes the AKT pathway by upregulating miR-21-5p (Figure 5). In addition, miR-21-5p-driven SMAD7 reduction leads the way to SMAD4 increase. It appears, therefore, that miR-21-5p stimulates osteogenic differentiation through at least two different routes. It is known that both the AKT and SMAD pathways upregulate the osteogenesis master gene RUNX2 expression.

## 4. Discussion

Recently, the regulatory functions of epigenetic factors such as miRNAs in physiological and pathological processes have been widely investigated. In particular, modulation of miRNAs following physical exercise represents an interesting field of investigation since these non-coding RNAs may be considered as defenders against degenerative diseases, as well as useful prognostic markers [15]. Different studies have shown that miRNAs can be recovered from peripheral blood and analyzed as biomarkers in several biomedical fields. In particular, previous studies show the modulation of circulating miRNA following exercise and chronic endurance. The upregulation of miR--338-3p, miR-330-3p, miR-223, miR-139-5p, miR-143 has been reported after 1 h of exercise [16]. Interestingly, it has been reported that miR-223 is involved in bone metabolism and in particular in osteoclasts and osteoblast differentiation [17]. Therefore, the upregulation of miR-223 following exercise suggests the induction of bone remodeling after physical activity. Moreover, increased expression of miR-21-5p and miR-378-5p in serum has been reported after acute exhaustive exercise [7]. Both miR-21-5p and miR-3785p play an important role in osteogenesis [5,7]. Accordingly, it has been reported that physical exercise plays a fundamental role in promoting mobilization of mesenchymal stem cells and their osteogenic commitment [9].

In this work we have demonstrated how the addition of POST RUN sera to cultured MSCs modulates the expression of miRNAs promoting osteogenic differentiation. Notably, the upregulation of miRNAs promoting osteogenic commitment of progenitors occurred at the expense of the mutually exclusive adipogenic commitment.

We monitored the cellular response to POST RUN sera exposure by measuring the expression levels of selected miRNAs. In particular, the expression of miR-21-5p, miR-129-5p, and miR-378-5p, were found to be upregulated in cells treated with POST RUN sera. The role of these miRNAs in promoting osteoblast differentiation has been broadly reported [5,7,18]. Li et al. showed that miR-21 promotes osteogenesis by inducing RUNX2 via the Smad7-Smad1/5/8 pathways [4]. Furthermore, it has been reported that miR-378 activates the PI3K/Akt pathway by targeting CASP3 [19], and miR-129 enhances RUNX2 expression by targeting STAT1, a transcriptional repressor of RUNX2 [6]. Our data show timing differences in their modulation during the osteogenic differentiation process of MSCs, monitored at the proliferative (7 days) and mineralization (14–21 days) stages.

We focused mainly on miR-21-5p’s modes of action, considering its persistent upregulation during the 21 days of differentiation period. MiR-21-5p is known to regulate cellular proliferation, invasion, and migration, by targeting the two TGF-beta pathway antagonists: PTEN and Smad7 [20]. MiR-21-5p’s positive role in promoting the osteogenic commitment of bone marrow stem cells has been demonstrated in vitro as well as in animal models [4,21,22]. Our experiments demonstrate that miR-21-5p upregulation induced by physical exercise, also promoted osteogenic differentiation through the AKT and SMAD pathways.

It has been suggested that increased levels of circulating miR-21 occur during physical exercise as a consequence of endothelial cells apoptosis following exercise [23]. In our study, we found that MSCs treated with POST RUN sera had a reduced tendency for apoptosis. This finding is in agreement with the described role of miR-21 in inhibiting apoptosis in mesenchymal stem cells via PTEN down-regulation [24]. Moreover, a study related to three protocols involving different muscular work reported that miR-21 levels decreased after the exercise, while increasing during the recovery. The authors suggest that miR-21 is involved in pro-inflammatory as well as in immunoregulatory anti-inflammatory processes [25].

Notably, we found that miR-188-5p expression was instead downregulated during the differentiation process of MSCs treated with POST RUN sera. The dowregulation of miR-188-5p is in agreement with the increased levels of AKT/pAKT observed in MSCs treated with POST RUN sera. In fact, it has been demonstrated that miR-188-5p reduces the PI3K/AKT signaling pathway [26,27]. MiR-188-5p has been suggested to play a regulatory role in promoting the adipogenesis versus osteogenesis switch [8]. In particular, it has been demonstrated that miR-188-5p contributes to bone ageing by increasing the number of adipocytes at the expense of osteoblasts. Indeed, the downregulation of miR-188-5p following physical exercise can be considered as a desirable effect counteracting age-related bone deterioration.

It has also been demonstrated that miR-129-5p inhibits preadipocyte proliferation by targeting G3BP1 (GTPase-activating protein SH3 domain-binding protein 1) [28]. Our data confirm that: miR-188-5p downregulation and miR-129-5p upregulation, as a consequence of physical exercise, lead to reduced expression of PPARG2, the master regulator of adipogenesis and to a reduced proliferation of adipocytes. Finally miR-378-5p upregulation not only promotes osteogenic differentiation [7], but is also involved in brown adipogenesis [29,30]. This finding is in agreement with the positive role of physical exercise in counteracting obesity. Therefore, we can speculate that physical exercise reduces adipogenesis in general and increases the brown fat component.

In conclusion, our work exemplifies the important regulatory roles played by a few selected miRNAs. They may be also considered very useful markers for the monitoring of osteogenic versus adipogenic commitment of progenitor cells in response to diverse environmental stimuli.

## Figures and Tables

**Figure 1 cells-08-00742-f001:**
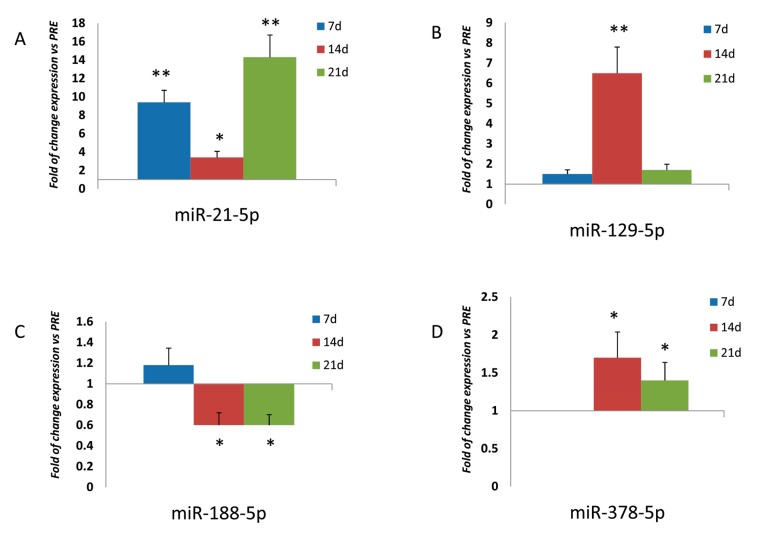
miR-21 expression was upregulated in MSCs treated with POST RUN sera during the differentiation (**A**); miR-129-5p was significantly upregulated only after 14 days of differentiation (**B**); MiR-188-5p was downregulated after 14 and 21 days of differentiation (**C**). In contrast, miR-378-5p resulted upregulated after 14 and 21 days of differentiation (**D**). Data were obtained from three independent experiments * *p* < 0.05 and ** *p* < 0.01.

**Figure 2 cells-08-00742-f002:**
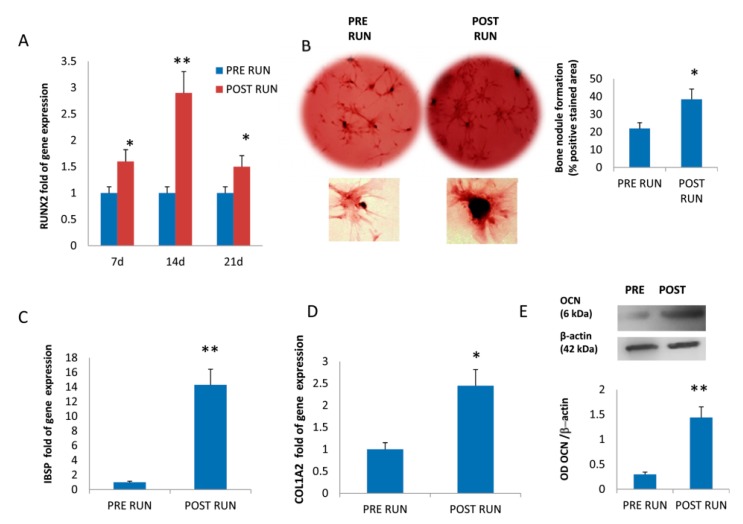
Upregulation of RUNX2 was observed in MSCs treated with POST RUN compared to PRE RUN sera (**A**). Bone nodule formation ability assessed by Alizarin Red S was in POST RUN sera MCSs (**B**). Increased osteogenic differentiationin in POST RUN sera MSCs was confirmed by the upregulation of Collagen type I alpha 2 chain (COL1A2; **C**) and Integrin Binding Sialoprotein (IBSP **D**) as well as by increased protein levels of Osteocalcin (OCN) (**E**) Data were collected from three independent experiments. OD: optical density; * *p* < 0.05 and ** *p* < 0.01.

**Figure 3 cells-08-00742-f003:**
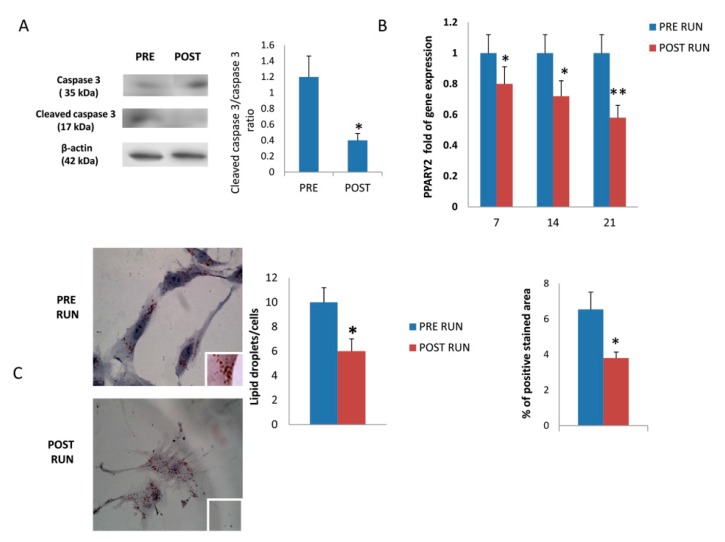
Cleaved caspase-3/caspase-3 ratio was reduced in MSCs treated with POST RUN sera (**A**) PPARG was downregulated in MSCs treated with POST RUN sera (**B**). The reduced adipogenic commitment was confirmed by a reduced number of lipid droplets for cells as well as the % of Oil Red O positive stained area (**C**). Data were collected from three independent experiments. * *p* < 0.05 and ** *p* < 0.01.

**Figure 4 cells-08-00742-f004:**
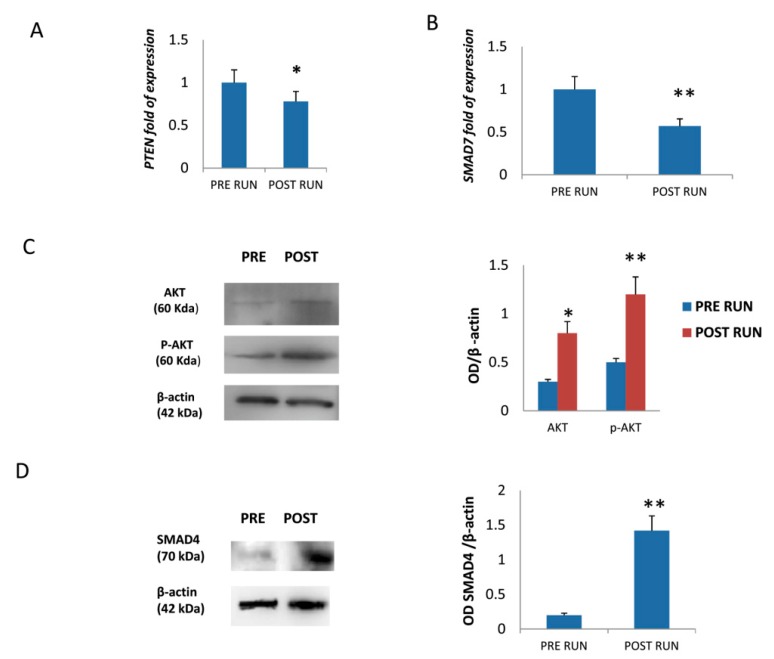
Downregulation of PTEN (**A**) and SMAD7 (**B**) mRNAs was observed in MSCs treated with POST RUN sera. Western Blot analyses showed increased levels of total AKT (**C**) as well as of SMAD4 proteins (**D**). Data were collected from three independent experiments. * *p* < 0.05 and ** *p* < 0.01.

**Figure 5 cells-08-00742-f005:**
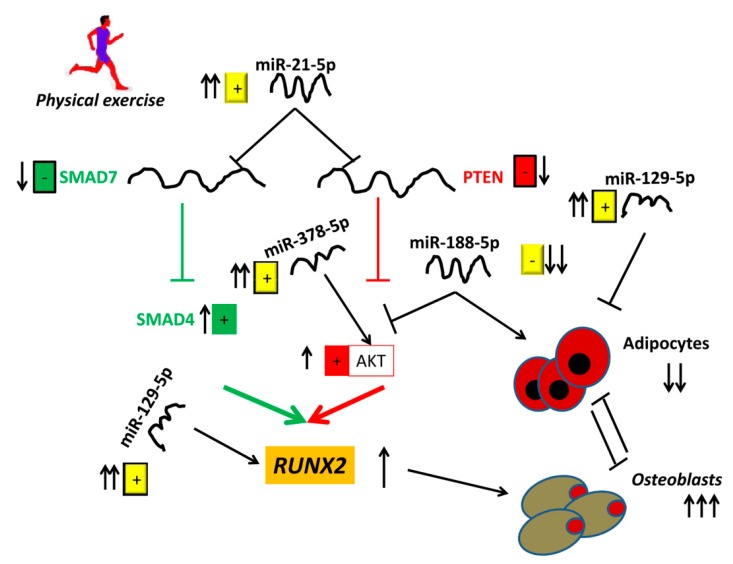
A schematic view describing the role of physical exercise in promoting osteogenic differentiation by inducing miR-21-5p upregulation.

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
