# Peer review of "Physical Exercise Modulates miR-21-5p, miR-129-5p, miR-378-5p, and miR-188-5p Expression in Progenitor Cells Promoting Osteogenesis"

_cells, 2019, doi:10.3390/cells8070742_

Round 1

Reviewer 1 Report

Valenti et al present a paper entitled "Physical Activity Modulates miR21, miR129-5p,miR378 and miR188-5p Expression in Progenitor Cells Promoting Osteogenesis". They recruited 20 male amateur runners participating to a half marathon, whose sera were collected before (PRE RUN) and after (POST RUN) the performance, were added to cultured human mesenchymal stem cells. Sera's effects were evaluated on the modulation of selected miRN and the consequent effects studied on osteogenic differentiation. There was an increase of miR21, miR129-5p, miR378,
levels and a decrease of miR188-5p levels.

The authors specifiy that they measured the levels of miR-129-5p and MiR-188-5p. What about miR-21 and miR-378? -5p or -3p?

Can the authors explain "In particular, stained areas were expressed as percentage respect to total area." ?

I would recommend to also measure the levels of miR-223 in this study as it was strongly involved in osteoclastogenesis.

Can the authors justify the use of RNU44 as a control gene?

It would be interesting to also measure the levels of the studied miRNAs in the serum of the patients to see the trend before and after exercice.

As explained in discussion, another study has shown that increased levels of circulating miR21 occur during physical activity as a consequence of endothelial cells apoptosis. So it would be interesting to assess apoptosis leves when cells are subjected to runner's sera per and post run.

The discussion is poor and there is not enough references. For exemple the article by Luo et al "MiRNA-21 mediates the antiangiogenic activity of metformin through targeting PTEN and SMAD7 expression and PI3K/AKT pathway" should be cited as they already showed that miR-21 targets PTEN and SMAD7. There are a lot of articles showing a role of miR-21 on bone formation that could be cited.

Minor please harmonise all miRNAs to miR-number as it is the international consensus. Why in the title did the authors use the 5p nomencalture for 2 miRNAs and not the others?

Reviewer 2 Report

Physical Activity Modulates miR21, miR129-5p, 2 miR378 and miR188-5p Expression in Progenitor Cells Promoting Osteogenesis

By Valenti et al.,

The authors cultivated mesenchymal stem cells with runners’ sera, collected before (PRE RUN) and after (POST RUN). They found an increased expression of miRNAs promoting osteogenic differentiation (miR21, miR129-5p, miR378, 19 respectively). miRNAs represent an emerging field of fine regulation of gene expression and the content oft he manuscript is novel since there involvement in bone remodelling post exercise is unknown. The manuscript is well written.

The question is why was collagen type I not tested? Some information is lacking in legends and some figures could be improved as mentioned below.

Line 16: „the performance“ sounds surplus

Line 17: mesenchymal stem cells, better to write „stromal“ instead of „stem“

Line 32: „bone making cells“, seems not exact since osteoblasts produce only ostoid, I would omit this explanation.

Line 33: „mature osteoblasts“ sounds also misleading

Line 41: why was collagen type I not tested as main protein component in bone?

Line 46: correct „mesenchimal“

Line 84: isobuthyl?

Line 104: there is a surplus blank

Figures

Figure 1 add „d“ for days in the diagrams. Place upper graph exactly in the position oft he lower one. Mention the number of independent experiments included in the legend (do this generally in each figure).

Fig. 2 explain the abbreviation for osteocalcin and „OD“ in the legend and explain what IBSP is. Bring the western blots in the same position. B: show also an overview at lower magnification. Insert „d“ for days at the x-axis.

Go through all figures and find a general writing style for β-actin.

Line 177: correct „evauated“

Figure 4: bring the graphs on the right side in a similar position.

Figure 5 should be improved it is sloppy arranged (distances between arrows and boxes, length and position of lines…)

Reviewer 3 Report

This manuscript demonstrates male runners show modulation of miR21, 129-5p, 378 and 188-5p expression in osteo progenitor cells. The authors have shown upregulation of miRs and in contrast  miR188-5p is downregulated following physical activity to promote osteogenesis. Though the manuscript is well written, I suggest the following changes or clarifications to further improve.

In general, the wording “Physical activity” can be replaced with “Physical exercise” to better reflect in the text and figures/legends.

Title- They may clarify using words “Physical exercise or Running induced changes…………. expression in osteogenesis progenitor cells”.

Abstract (2nd line)- correct the words “short non-coding RNAs”

Results (line 144) clarify “3.1. Expression of miRNAs in osteoprogenitor cells” as the miRNAs analyzed are not specific to bone forming cells.

Line: 172; clarify the word “---- PPARγ2 expression (Figure 3A), Fig.3 legend and Abbreviations (line 262).

The authors did not show data related to increased osteoblast numbers/bone nodule formation.

It is unclear why the authors did not note other miRs other than miR21 upregulation shown in Fig.5 illustration to describe the role of physical activity in promoting osteogenesis.

Discussion (lines 231-232)-  The statement that “downregulation of miR188-5p following physical activity can be considered as a desirable effect counteracting age-related bone deterioration” is not illustrated in Fig.5.   

Round 2

Reviewer 1 Report

changes are satisfactory

Reviewer 3 Report

p.3; 138   Please correct GAPDH in   "genes (β₂microglobulin and GAPDH...….."